# Towards Convergence: How to Do Transdisciplinary Environmental Health Disparities Research

**DOI:** 10.3390/ijerph17072303

**Published:** 2020-03-29

**Authors:** Clare E.B. Cannon

**Affiliations:** Department of Human Ecology, University of California, Davis, CA 95616, USA; cebcannon@ucdavis.edu

**Keywords:** transdisciplinary, health disparities, social vulnerability, environmental health, social sciences, team science, convergence research, community-based participatory research

## Abstract

Increasingly, funders (i.e., national, public funders, such as the National Institutes of Health and National Science Foundation in the U.S.) and scholars agree that single disciplines are ill equipped to study the pressing social, health, and environmental problems we face alone, particularly environmental exposures, increasing health disparities, and climate change. To better understand these pressing social problems, funders and scholars have advocated for transdisciplinary approaches in order to harness the analytical power of diverse and multiple disciplines to tackle these problems and improve our understanding. However, few studies look into how to conduct such research. To this end, this article provides a review of transdisciplinary science, particularly as it relates to environmental research and public health. To further the field, this article provides in-depth information on how to conduct transdisciplinary research. Using the case of a transdisciplinary, community-based, participatory action, environmental health disparities study in California’s Central Valley provides an in-depth look at how to do transdisciplinary research. Working with researchers from the fields of social sciences, public health, biological engineering, and land, air, and water resources, this study aims to answer community residents’ questions related to the health disparities they face due to environmental exposure. Through this case study, I articulate not only the logistics of how to conduct transdisciplinary research but also the logics. The implications for transdisciplinary methodologies in health disparity research are further discussed, particularly in the context of team science and convergence science.

## 1. Introduction

Recent scholarship increasingly recognizes the importance of multi-disciplinary perspectives that also seek to leverage diverse approaches and experiences to integrate (interdisciplinary) and transcend (transdisciplinary) disciplinary bounds to advance scientific knowledge to tackle pressing social problems (e.g., [1,2,3,4,5,6,7,8,9,10]). Important national public funding agencies, including the National Institutes of Health (NIH) in the U.S., a collection of 27 institutes and centers that study and fund a broad portfolio of health and health processes, have also begun to drive relevant fields towards integratory and transcendent research through their funding calls for research proposals and centers, as outlined in the NIH Roadmap for 21st Century Medical Research [8,11]. The NIH is one of the primary funders of health and health disparities research, investing $39.2 billion annually [12] and supporting roughly 83% of federally funded biomedical research [13]. The NIH is increasingly recognizing and funding Team Science approaches—leveraging diverse approaches and experiences from different fields [14]—to solve the most complex biomedical and public health questions of our time. Similarly, the U.S. National Science Foundation (NSF) spends $7.8 billion annually and funds approximately 24% of all federally supported basic research conducted by U.S. universities [15]. Since 2016, the NSF has committed to growing convergence science—"research driven by a specific or compelling problem and a deep integration across disciplines” [16]—as one of its ten big ideas or top funding priorities. The commitment of both public funding agencies to the cross-fertilization and collaboration of researchers from multiple disciplines heralds the increasing importance that these types of research collaboratives will have in studying a wide range of social, ecological, and health concerns. While federal funding agencies incentivize such collaborative work, barriers and challenges remain.

To better understand the benefits and challenges to transdisciplinary research, I review the latest transdisciplinary scholarship in health disparities and environmental research. Next, I analyze a case study that uses a transdisciplinary approach to researching environmental health disparities, paying close attention to interactions between built and social environments. This case study is analyzed to provide further insight into (1) how to do transdisciplinary research; and, (2) to identify the specific challenges of doing such research. Finally, through a synthesis of the literature and lessons learned through the case study, the paper identifies barriers at the institutional level with specific recommendations to overcome these barriers.

## 2. Literature Review

There are three main types of cross-disciplinary research: multidisciplinary, interdisciplinary, and transdisciplinary approaches. Multidisciplinary research typically includes scientists from many disciplines working together at some point in the research process, formulating and addressing separate research questions [1,17]. Interdisciplinary approaches attempt to transfer knowledge from one discipline to another, which may, in turn, create an entirely new discipline (i.e., health geography). Through integrating work from different disciplines, knowledge may not be as effectively communicated back to specific discipline structures, thereby missing opportunities to advance knowledge within already codified disciplines [17,18]. For example, Gehlert et al. argue that health disparities, in particular, are hard to shed light on through interdisciplinary work due to the complex interactions at multiple scales of analysis (i.e., biological, behavioral, social, and environmental) [17]. Alternatively, transdisciplinary approaches attempt to form their own intellectual space by understanding the world in its complexity through a novel paradigm of collaboration that both transcends disciplinary bounds in the creation of new knowledge and integrates knowledge to effectively communicate within disciplinary structures [1]. Transdisciplinary approaches are “primarily a process of assembling and mapping the possible interconnections of disciplinary knowledge about any given…problem until the fullest possible understanding of the problem emerges” [10] (p. 1560). Such transdisciplinary approaches may help legitimize different ways of knowing and reduce gaps in knowledge systematically [19]. Moreover, transdisciplinary approaches offer an advantage to research into health disparities as they bring together experts from multiple, relevant disciplines to work within and outside of traditional boundaries to address multilevel determinants of health disparities and their interactions [17]. The current paper focuses on transdisciplinary approaches, particularly in public health and environmental research.

### 2.1. Understanding Transdisciplinary Approaches

Transdisciplinary research has the potential to provide a systematic and comprehensive framework for defining and analyzing multiple dimensions (e.g., social, economic, political, environmental) of factors scalarly that influence human health and the environment [1]. Transdisciplinary research tends to be characterized by four key traits, (1) a relationship with socially relevant issues; (2) transcendence of disciplinary paradigms; (3) a participatory research design; and, (4) a search for unifying knowledge [2]. Transdisciplinary approaches lend themselves to the study of socio-ecological problems, such as climate change, natural hazards, and pollution, through the identification of the interlocking and multiple causes and consequences of these problems [2,10,17,20,21]. Given the importance of transdisciplinary research in advancing knowledge and as a priority research area for funding, the relative paucity of research into the methodological and theoretical aspects of transdisciplinary research [3,22,23,24] and the practice of transdisciplinary scholarship itself is surprising [4,25,26].

Important epistemological contributions have been made by transdisciplinary scholars through their own research [2,4,25,27]. There are three major guiding considerations identified through this literature. First, transdisciplinary research requires increased coordination across diverse research personnel [4]. Building on previous case studies of scientific collaborations for health promotion (e.g., [28,29]), Stokols stresses the importance of coordination with both community–researcher partnerships in action research and the coordination necessary to produce transdisciplinary science [4]. However, the similarities and differences among these different levels of coordination among action research partners and transdisciplinary science partners has yet to be studied [4]. Second, transdisciplinary research necessitates collaboration across different discipline-specific jargon, methodologies, and topical area priorities [2,4]. Third, transdisciplinary approaches facilitate research across ecological levels (i.e., cell to society) and may more readily increase the goals of translational research (from basic science to communities) [2].

Although there are clear benefits to transdisciplinary research, there are several key elements which also pose challenges to conducting this kind of research, as identified in the extant literature. Table 1 summarizes the key elements, strategies and challenges of doing transdisciplinary research, as identified in the extant literature. First, work requires an ethic of openness and respect towards multiple and different perspectives (e.g., [4,5,7,8,25,27,30]). Second, bridging different discipline boundaries requires what has been termed a “boundary-spanner.” This figure is essential for navigating the conflicts and tensions that arise between different perspectives and team members and to build trust [27]. The leadership of the boundary-spanner is found to be integral to the success of these collaboratives (e.g., [7]). Third, team members must be flexible enough to allow for multiple pathways of integration and collaboration across discipline norms, frameworks, and boundaries [5]. Through such collaborative research, new epistemic communities must be formed, and new kinds of evidence produced [5].

Fourth, mutual confidence and trust with a commitment to mutual learning are necessary to undertake collaborative research [6]. For instance, Harris and Lyon found that trust is essential in successful collaborations, particularly since researchers are opening themselves up to vulnerability and risk, and have multiple competing values and objectives and increased uncertainty [27]. They found four key means of building trust. First, trust is built by having information on collaborators (bios, websites, pictures, CVs, etc.). Second, trust is built through prior experiences of working together. Third, all team members need a clear understanding of norms for collaboration. And, fourth, there need to be guarantors of relationships [27]. Fifth, transdisciplinary research must learn how to communicate across various discipline-specific languages [6] necessitating and creating what some have termed a “safe space” [31]. Sixth, the complexity, multiplicity, and contextual specificity of transdisciplinary research requires a stability across expertise and subjectivity as well as between interdisciplinary integrations and disciplinary specificity [5]. Similarly, Harris and Lyon argue that the norms of monodiscplinary research must be changed to understand how diverse teams can be created and sustained [27]. Lastly, studying the world in this complexity provides a good opportunity to make the best possible decisions given the inevitable uncertainty in an imperfect world [6].

To do transdisciplinary research there are several key strategies that facilitate the development of effective research teams [9]. They are, (1) institutional support for the approach; (2) diverse team members; (3) cross-disciplinary training and opportunities for shared problem solving; (4) shared language and goals in operationalizing the research; and (5) multidirectional communication. Lastly, this kind of scholarship requires a willingness to commit substantial time to collaboration; an openness to learn other disciplinary languages and cultures; and the capacity to build trust and confidence [8]. Understanding what advances and hinders collaborative research is necessary in order to better promote and support collaboration.

Accompanying these key elements to doing transdisciplinary research, there are five significant challenges. First, these kinds of collaboratives are very labor intensive and their potential scientific and community benefits may not be clear for many years [4]. Second, features particular to this kind of research, such as “the gap between conventional metrics and the complexity of transdisciplinary research,” make it difficult to evaluate [5]. Third, researchers tend to have more reasons for non-collaboration than reasons to collaborate [25]. For instance, in his analyses of interviews with researchers, Pohl (2005) found that collaboration evolves in a problem-driven research environment and tends to flow along the lines of division of labor [25]. Scholarship on the subject has found that the pressure to produce usable results—as defined in relationship to a disciplines’ norms and values—needs to be reduced in order to increase collaboration (e.g., [1,25,32]). Fourth, in their study on building trust across research collaborations in ten case studies, Harris and Lyon (2013) identified disincentives to doing this kind of research, including single-disciplinary publications, and fears that research will not be perceived by any of the disciplinary communities as rigorous enough [27]. Fifth, Kueffer et al. (2008) argue that academic publishing is a significant barrier to doing and promoting transdisciplinary scholarship [32]. They argue for the need to establish transdisciplinary research journals and for discipline specific journals to provide special issues and sections dedicated to transdisciplinary research to reduce this barrier.

Finally, similar to other forms of transdisciplinary scholarship, challenges to doing transdisciplinary research include difficulties in assigning roles for team members across levels of expertise and rank; the necessity of not defining the problem narrowly or too broadly; and the necessity of overcoming discipline specificity and rigidity [8]. One important obstacle is that universities themselves can be a barrier to transdisciplinary research through their lack of support, both in terms of funding and in performance evaluation (i.e., advancement, tenure and promotion) [33].

Transdisciplinary approaches, including team science in the health sciences, convergence research in basic science, and collaborative scholarship in the humanities, have sought to transform the ways we understand and solve our most pressing socio-ecological problems. In particular, transdisciplinary research in public health has sought to reframe the problems that produce and sustain health disparities in order to more quickly alleviate them. To this end, the current extant literature in transdisciplinary research in public health and health disparities is reviewed. The opportunities for this kind of research and the transformative science it makes possible is evidenced by NIH support and the subsequent promotion and sustainability of research collaborations across the health and social sciences [34].

### 2.2. Transdisciplinary Approaches to Public Health and Health Disparities Research

Public health scholars have argued for the importance of transdisciplinary approaches to framing and understanding interactions among built and social environments and health outcomes generally, and health disparities and environmental health specifically [8,10,17,26]. These approaches include translational science [9], public health exposome [20], Team Science [10], One Health [35], and adaptive and participatory [36]. For instance, Dankwa-Mullan and colleagues argue that transdisciplinary health disparities research applies an integrative approach to solving health disparities that is not only translational but also transformational [9]. Such an approach takes into account structural inequalities and provides a foundation for innovations that will lead to practical implementation in communities. This kind of research can also assist the translation of basic and clinical science into more effective health and environmental policies [20]. Translational, transformational, and transdisciplinary research is necessary to tackle the inherent complexities of the social problems generated by climate change and health disparities.

Given the complex and varied causes of health disparities, Juarez et al. (2014) similarly argue that new approaches are needed to reduce health disparities given the lack of meaningful progress in these inequalities [20]. They argue that current approaches have not adequately related the complexity of relationships among environment, personal health, and population level disparities. For instance, public health exposome uses a socio-ecological approach to exposomes to create an exposure-tracking framework to integrate the complex relationships between exogenous and endogenous exposures across the lifespan [20]. Moreover, understanding environmental science and public health exposomes presents an opportunity to train a new generation of transdisciplinary scholars.

Similarly, scholars have begun to employ transdisciplinary approaches in environmental health research (e.g., [7,21,35]). With the increasing complexity and urgency wrought by the climate crisis, Team Science, specifically, and transdisciplinary research, broadly, are key to framing and analyzing issues of health and sustainability to generate solutions of mitigation and adaptation [10]. Another example of transdisciplinary approaches across socio-ecological levels is the One Health framework. Min and colleagues (2013) argue that One Health approaches must transcend disciplinary boundaries [35]. One form of health research aims to solve complex health challenges at the animal–human–ecosystem interface by investigating the social, physical, and environmental determinants of health (e.g., [6,37]).

Additionally, scholars have begun calling for adaptive, participatory, and transdisciplinary approaches to solve the so-called wicked or messy problems of cumulative impacts, or the range of impacts due to environmental hazard and social vulnerability [36,38]. For instance, Shrestha and colleagues (2018) argue for the importance of cumulative burden assessment (CuBA) as a tool to inform planning and decision making on health disparities related to multiple environmental burdens [21]. However, to account for social complexity, CuBAs require adaptive, participatory, and transdisciplinary approaches are necessary. Importantly for reducing disparities, CuBAs can be used to determine distributional environmental justice issues (i.e., an uneven distribution of environmental burdens) and related adverse health outcomes to inform planning and decision-making [21].

Given the challenges to transdisciplinary research elaborated above, there are several resources that may help facilitate transdisciplinary research in the health sciences. One such mechanism is through transdisciplinary promoters, such as through research centers or “accelerators.” For example, Horowitz and colleagues (2017) argue for “accelerators”, rooted in Team Science, to foster collaborations to generate new ideas, questions, and approaches [39]. They argue that health disparity interventions have been inadequate, since research traditionally takes place in “disciplinary, disease, and demographic silos”, and that team science for translational research could potentially overcome these barriers. Another example of a transdisciplinary research promoter is evidenced by the Collaborative Research Center for American Indian Health (CRCAIH), which aims to build tribal research infrastructure and increase transdisciplinary research in American Indian/Alaska Native (AI/AN) health [37]. Through pilot grants to promising transdisciplinary teams, the CRCAIH focuses on the social determinants of health and health disparities experienced by AI/AN in its regional area (South Dakota, North Dakota, and Minnesota). Similarly, Holmes et al. (2008) also leveraged a multilevel health disparities research center to use transdisciplinary approaches to study the social determinants of health disparities [26].

Social science also has a vital role to play in facilitating transdisciplinary health disparities research. Similar to Stokols’ (2006) concept of transdisciplinary action research [14], Cordner and colleagues (2019) argue for combining social science with environmental health research and situating it within a larger constellation of transdisciplinary research [7]. They argue that social science and environmental health collaborations offers the greatest potential for improving public and environmental health. Although generally there is a lack of infrastructure to support translational team science, research teams can build a community of practice [39]. These communities of practice could inform how we organize and structure research training cores with different kinds of research centers (i.e., P30 NIEHS, P50 NIMHD) [20]. Additionally, training programs in transdisciplinary approaches and experience doing such research is an important resource for future iterations of transdisciplinary research. Critically, the support from departments and institutions informs the degree of collaboration between social science and health researchers [7]. Subsequently, training programs are very important for increasing transdisciplinary communication and to consciously develop a practice of collaborative research [7,40].

Drawn from the extant literature, there are several key elements to *doing* transdisciplinary research into health disparities and environmental health. Black and Black (2009) along with Higginbotham and colleagues (2001) identified six main steps to doing transdisciplinary work [10,41]. These include (1) defining the problem; (2) assembling a team of researchers; (3) reviewing existing disciplinary and interdisciplinary knowledge; (4) research design based on the review; (5) refining conceptual understandings; analyzing data using strategies from multiple disciplines; and (6) recommending interventions to resolve the problem. Similarly, in their scoping review of one piece of health literature, Min et al. 2018 found nine emerging themes in transdisciplinary research: (1) education, (2) discipline conflict, (3) meaningful communication, (4) collaborative conceptual framework, (5) guidance, (6) perceived power differences, (7) community-based approaches, (8) support for transdisciplinary research, and, (9) time- and effort-intensity [35] (p. 26).

Several transdisciplinary frameworks for advancing health sciences have been presented here. Another important approach for understanding health disparities at multiple socio-ecological levels is transdisciplinary action research. Transdisciplinary action research joins action research, a collaborative approach to inquiry with a focus on implementing problem-solving actions usually within a community context, with transdisciplinary approaches to create a novel science to increase its own sustainability and achieve better outcomes [4]. Such action-oriented transdisciplinary research is increasingly necessary in order to move analyses from molecular levels to community and policy perspectives on the problems (climate crisis, health disparities, etc.) we seek to understand and solve. To this end, this article fills a gap in the scientific literature regarding how to do transdisciplinary research, with recommendations for evaluating such research given its various forms and levels of collaboration and the interaction of these forms [29]. The current article aims to build on the foundational literature reviewed here through a case study that uses a transdisciplinary approach to community-based participatory action research (CBPAR) to better understand the potential routes of environmental exposure for a small, rural disadvantaged community in California’s San Joaquin Valley. Next, the best practices and lessons learned from this case study are elaborated. Finally, recommendations on reducing barriers to this kind of research, particularly at the university-level, are discussed.

**Table 1 ijerph-17-02303-t001:** Key elements and descriptions, strategies, and challenges for understanding and doing transdisciplinary research with major studies from the extant literature.

Key Elements	Description of Key Elements	Important Strategies	Significant Challenges	Major Studies in the Field
Openness and respect	Ethic of openness and respect towards multiple perspectives	Institutional support for transdisciplinary approaches	Labor and time-intensive	Stokols, 2006 [4]; Dankwa et al., 2010 [9]
Boundary-spanner	Boundary-spanner to bridge different discipline boundaries	Diverse team members	Difficult to evaluate	Harris & Lyon, 2013 [27]; Cordner et al., 2019 [7]
Flexibility	Flexibility to allow multiple pathways of integration and collaboration across discipline norms, frameworks and boundaries	Cross-disciplinary training and opportunities for shared problem solving	Disincentives including fear that research will not be perceived by discipline-specific communities as rigorous enough	Pohl, 2005 [25]; 2010 [2]
Confidence and Trust	Mutual confidence and trust with a commitment to mutual learning	Capacity to build trust and confidence	More reasons for non-collaboration than collaboration	Annerstedt, 2010 [6]; Gehlert et al., 2010 [17]
Communication	Communication across various discipline-specific languages	Shared language and goals in operationalizing the research	Academic publishing organized around disciplines	Black and Black, 2009 [10]; Pereira et al., 2015 [31]
Stability	Stability across expertise and subjectivity	Make and invest time to build collaborations	Difficulty in assigning roles to team members	Klein, 2008 [5]; Horowitz et al., 2017 [39]
Complexity	Complexity that provides the opportunity to make best possible decisions given uncertainty in an imperfect world	Understanding what advances and hinders collaborative research to support and promote collaboration	The need to not define the problem of analysis too narrowly or broadly	Rosenfield & Kessel, 2008 [8]; Shrestha et al., 2018 [21]

## 3. The Case Study

The case is a pilot study designed to measure exposure in a sample of residents from a small, rural, unincorporated township in California’s San Joaquin Valley. The study illustrates the key characteristics, including transcending multiple disciplinary boundaries, the time-intensive nature of the project, and trust-building, elucidated above. It provides valuable insights and strategies on *how* to do transdisciplinary research in the study of environmental health disparities. This section is organized as follows: the case study rationale and approach are explained; the research team including scientists from multiple disciplines and community partners is described; and strategies for organizing and conducting transdisciplinary research are enumerated. The analyses and final results of collected data are not discussed here due to space limitations and the current paper’s focus on doing transdisciplinary research.

### 3.1. Interaction of Social and Environmental Contexts

For this study, we measured environmental exposures and related health outcomes for a sample or residents from Kettleman City, a predominantly Spanish-speaking, agricultural unincorporated township in California’s San Joaquin Valley. The research followed all ethical guidelines for research with human subjects and has IRB approval. One of the founding sites of environmental justice due to their successful fight against the siting of a waste incinerator near the hazardous waste landfill there [42], Kettleman City continues to be an important site of study to understand both the unique effects of hazardous waste landfills on human health as well as the cumulative impacts of other pollutant sources. The built environment of the town means that the approximately 1500 residents there may be disproportionately exposed to increasing amounts of pollution. The town lies at the intersection of two major highways, I-5 and CA-41, surrounded by industrial agriculture, primarily almond and pistachio groves (see Figure 1). The community is also approximately two miles from one of two operating class I hazardous waste landfills in the state of California (the Kettleman Hills Landfill). Class I hazardous waste landfills contain the most severe hazardous waste that can be disposed of, including waste from oil production processes [43].

Other pollutant sources include particulates and volatile organic compounds (VOCs) from air pollution, particularly diesel trucks from traffic, as well as contaminated water with elevated concentrations of benzene and arsenic that exceed state standards [44]. Moreover, such health effects may be compounded by social vulnerability—those social factors and inequalities that influence the susceptibility of various social groups to harm and that govern such groups’ ability to respond [45]. The experiences of both social and environmental stressors, more broadly, have been found to potentiate adverse health outcomes such as higher rates of asthma, cancer, and diabetes, compared to those in the U.S. not exposed to such pollution [46]. Subsequently, rural populations in the Valley may face a ‘dual risk’ of heightened exposure to environmental and social risks, with few regulatory resources to cope with these challenges.

Similar to many in the San Joaquin Valley, Kettleman City is a low-income community with low access to healthy food, including fresh fruits and vegetables [47]. Although the Valley produces 8% of the U.S.’s agricultural output by value, generating roughly $50.13 billion annually [48], there is an average 12.8% of food insecurity, meaning approximately 550,000 people do not have access to enough affordable and nutritious food [49]. According to California Enviroscreen 3.0, a state-funded data visualization tool that compiles both social and environmental metrics, Kettleman City is located in a census tract in the 85%–90% percentile for social vulnerability and environmental degradation [50]. As a community that hosts a hazardous waste landfill and as an unincorporated township in California’s Central Valley, it is important to understand the myriad exposures community members face in order to determine what may contribute to their reported adverse health outcomes. Such information can advance our understanding of the intersections of the environment, exposures, health, and social inequality as well as assist policymakers in designing more equitable policies, and to provide community advocates with the latest, accurate scientific information.

This study is funded as a pilot project by the UC Davis Environmental Health Sciences Core Center (EHS CC P30ES023513; EHSC), a Core Center funded by the National Institutes of Environmental Health Sciences (NIEHS) of the U.S. NIH. Similar to other kinds of national public funding for research, the NIEHS funds Core Centers to centralize scientific resources and facilities to advance scientific research, facilitate team science, promote community engagement, advance translational research, and support future researchers [51]. Each center has an overall strategic research vision and includes four cores: Administrative, Integrated Health Sciences Facility Core, Community Engagement Core, and other optional cores [51]. To understand residents’ risk to multiple sources of pollution and social vulnerability, a community-based transdisciplinary approach was necessary to conduct this research.

### 3.2. Study Research Design

To this end, the current study was designed using a transdisciplinary, CBPAR approach. CBPAR is a kind of research in which scientists partner with community organizations in order to do rigorous and relevant research with an extended reach in science, the community, and decisionmakers [52]. For this study, scientists worked with El Pueblo Para el Aire y Agua Limpia (People for Clean Air and Water) of Kettleman City (El Pueblo) and Greenaction for Health and Environmental Justice (Greenaction), two environmental justice organizations with extensive and long ties to the community. These community organizations were partners throughout the entire research project. Helping to design the study, community partners identified major environmental stressors of concern together with the Principle Investigator (PI), and working with multiple scientists across a range of fields, including mechanical engineering, biomedical engineering, public health, biostatistics, policy, and chemistry. Bringing together a collaborative team from across campus, these transdisciplinary projects have been shown to advance the aims of both environmental health and justice [53]. For example, community partners, some of whom are lifelong residents of the town, described smelling pesticides sprayed on neighboring orchards, and had previously counted over 400 trucks per day on CA-41 heading to the landfill. Although most residents do not drink the tap water due to its known contamination with arsenic and benzene [44], were also concerned that they may be exposed to heavy metals and other elements in the water through cooking, cleaning, and bathing. To address these concerns, the study design incorporates collection of five types of data: air samples (i.e., particulates and VOCs), water samples, biological samples, and a community health survey of all households (see Table 2).

In collaboration with the research team, including community partners, the study employed a sequential multi-method approach. A community environmental health survey was developed and deployed first in order to collect data on the environmental stressors identified by residents. The second phase of the study sought to collect relevant environmental and biological data on these environmental stressors. Data collection methods are elaborated below. In order to design and conduct the study, the team members, including all research scientists and community partners, had to adopt an attitude of openness and respect [4], be flexible in applying their disciplinary approaches and community perspectives [25], build confidence and trust in one another [6], and develop multiple kinds of communication strategies [31] (see Appendix A, Figure A1, a flowchart for doing transdisciplinary research.)

In the fall of 2018, the PI, with two undergraduate research assistants and a graduate research assistant, went door to door delivering an 86-item survey to every household in the town (*N* = 300). The survey was developed with community partners, two residents, and reviewed by an evaluation expert and based on similar CBPAR environmental health surveys [54,55]. There were questions regarding health outcomes, such as if anyone in the household had asthma, or if anyone in the household had cancer, and routes of potential environmental exposure, such as if they smelled pesticides at their home, and if they used municipal water for bathing and cooking. (*n*_Responses_ = 45 households) (see Appendix A, Table A1 and Table A2, for relevant survey questions and responses). Although low, this response rate is on par with similar kinds of studies in rural, low-income communities [56,57]. Given that this was a pilot study, there were not funds for renumeration for survey participation, which would most likely have increased survey participation. Working with two organizations with extensive and long ties to the community, along with this survey and a review of secondary data sources (i.e., California Water Report and CalEnviroscreen 3.0), were a primary means of identifying residents’ health concerns.

Once community members identified environmental stressors of concern through the survey as well as through a review of secondary data, working with a transdisciplinary team, we set out to determine the best means of identifying and measuring target analytes to address these concerns (see Table 2). For example, we needed to identify potential effects due to the extensive traffic around the town. To do this, we needed to measure diesel particles, a carcinogen in the state of California that has been linked to adverse health outcomes [58]. To measure air pollution, the PI worked with an atmospheric scientist, a research and development engineer, two mechanical engineers, a chemist, and two public health scientists. The final research design included a trailer parked in the town for two weeks with four PM_2.5_ Interagency Monitoring of Protected Visual Environments (IMPROVE) monitors [59] and four custom-built micro gas preconcentrators to monitor VOCs [60] sampled over the course of a 12-day period.

A purposive snowball sampling strategy was used to generate a cohort of ten residents, five men and five women, with ages ranging from 20 to 81. Given that this is a pilot study, we aimed to assemble a diverse group of residents to collect samples from. The study cohort provided both household water samples and biological samples. For residents’ water contamination concerns, since we already had measured concentrations of benzene, chlorine, and heavy metals from the state monitoring report [44], we decided to focus our efforts on identifying and measuring concentrations of trihalomethanes (THMs) and haloacetic acids (HAAs), which are byproducts that form from the use of chlorine to kill chloroform bacteria in municipal water supplies. THMs and HAAs have been linked to carcinogenity, hepatotoxicity, nephrotoxicity, and endocrine toxicity [61]. Working with a geologist, chemist, and research engineer, the PI collected water samples from each study member’s house to identify and measure THMs and HAAs.

Lastly, residents reported a concern about polychlorinated biphenyls (PCBs) due to previous improper handling at the hazardous waste facility [62]. Four public health scientists, a biostatistician, and the PI worked together to design this aspect of the study to identify target analytes for PCBs to address residents’ concerns. A phlebotomist from the university was hired and traveled to the town to collect blood samples from the study cohort.

## 4. Discussion

### 4.1. Keys to Success

There are several key elements that contributed to the success of this research and that illustrate important considerations for doing transdisciplinary research. These elements include the social-scientist-as-boundary-spanner, regular contact with team members, and the support of a federally funded center. First, the PI, as a social scientist, was uniquely situated to be a “boundary-spanner” (Harris and Lyon, 2013) bridging the research scientists and community partners [27]. The PI, as a social scientist, has extensive research experience working with community organizations on socially and policy relevant problems, particularly around socio-environmental inequality. Similar to previous research [7], training from the social sciences helped the project succeed in that the PI was able to work across community organizations and with team members to achieve the project objectives. Moreover, the PI was well-situated in the social sciences to advance environmental and health disparities research and to operate as a boundary-spanner for the necessary coordination of the research personnel to achieve the study aims. This boundary-spanning is illustrative of the coordination across community partners and cross-disciplinary research partners necessary for transdisciplinary research [4]. Second, the PI, as a social scientist, was well-situated to communicate across community organizations and with biophysical, natural, and public health scientists on the team. As Harris and Lyon (2013) found in their analysis, transdisciplinary research opens researchers up to greater risk and vulnerability as they pursue studies outside of their immediate disciplines [1]. As a boundary-spanner on the project, the PI was able to build trust among diverse project participants through on-going engagement with project stakeholders. Given the need for coordination and expertise in CBPAR, the PI devised a monthly communique to community partners to discuss outstanding matters, identify and troubleshoot problems, and give project updates. This communique took the forms of in-person meetings, conference calls, and email briefs and reports. This monthly communique was essential in building trust as well as overcoming a perceived researcher silence that is often reported by partnered community organizations that do not hear from researchers for months on end [27]. This temporal disconnect speaks to the differing time horizons of community organizations and researchers, in which the former is often tied to decision-making deadlines in their advocacy efforts and the latter might take years to collect, analyze, and publish study results. Lastly, the PI traveled to the research site regularly (at least once every quarter) to meet with community partners, residents, and conduct fieldwork. As others have shown (e.g., [63]), in-person meetings and phone calls tend to be the best communication pathways with community organizations, as email is not always the best means of connecting. This hands-on work was vital to building trust with residents and community partners and highlighted the commitment and dedication of the researchers to addressing residents’ environmental health concerns.

Monthly communication was an important tool in maintaining coordination with community organizations and communicating even if no updates had occurred on the project (due, e.g., to lab availability and other demands on researchers’ time such as teaching, service, and other projects). The monthly report also served as a tool to more fully incorporate community organizations into the CBPAR approach and decide together how to troubleshoot the problems that arose and to advance project aims. This communique also proved useful in developing multiple kinds of research products for both non-academic audiences (i.e., research briefs, report back to participants, policy briefs), and for academic audiences (i.e., presentations, publications, and reports to funding agencies).

Additionally, coordination with community partners helped to inform coordination efforts with research team members [4]. The PI also coordinated monthly with the transdisciplinary scientific research team members in order to troubleshoot any issues that arose. These monthly communications occurred as in-person meetings, conference calls, and emails. The topics discussed included research design, data collection, data analysis, and interpreting findings.

Lastly, the federally funded EHSC, which funded the case study, also provided the PI access to its network of researchers. As an on-campus center designed to promote community-engaged research, the EHSC demonstrated a commitment to the PI, through mentorship, and to the project, through core directors’ participation on the study team. Taken together, the research team included fourteen scientists, including the PI and two community organizations. Similar to other transdisciplinary studies cited above (e.g., [7,37,39,64]), this center was instrumental in facilitating connections among the research team. Since the center had already built a network of multi-disciplinary researchers across the university, the PI was able to successfully leverage this to assemble a transdisciplinary research team to conduct this study. The on-campus center’s focus on community engagement meant that a network of community organizations was already established. These connections were equally important in facilitating initial connections and advancing trust with community partners in this study.

### 4.2. Elements of TD Research: Challenges as Opportunities

The team was transformed by one of scientists from multiple disciplines and community organizers into a transdisciplinary scientific team doing what Stokols (2006) termed “transdisciplinary action research”—the linking of community-engaged research and collaborative science [4] (p. 64). The transdisciplinary approach was not simply the coming together of people with diverse backgrounds and skillsets. Rather, team members came together to transcend discipline boundaries to understand which of the environmental and social stressors experienced by residents may impact their health. As described above, transdisciplinary research often both transcends disciplinary boundaries to think differently about the research subject and speaks within a disciplinary field (sociology, public health, environmental studies, etc.) [8]. The case study illustrates how problem solving was as an important site of *doing*—building trust, designing the study, data collection and analysis, and dissemination—transdisciplinary research.

One significant mechanism for conducting transdisciplinary research was through collaborative problem-solving. For instance, once we had the equipment, we needed a central place in town from which to sample. In working with local community partners, we decided it would be better to set up the monitors in a public area so that residents would not have to worry about the equipment being damaged or stolen. We thought that securing the monitors to a roof would get us the best data, while also ensuring the safety of the equipment. We investigated two promising central places that would presumably be safe: the local sheriff substation and local elementary school. We simultaneously contacted the local county sheriff and the school administrators. The school administrators did not return any emails or voice messages. On closer inspection of Google Maps satellite images, the pitched angle of the sheriff substation’s roof made it impossible to set up the monitors there. Since the roof was inaccessible, we decided that placing the monitors on a large wooden structure would allow us to get high quality samples. Between the substation and the local library, there was a huge grass yard. We endeavored to set up the monitors on the wooden stand there and power them from the sheriff substation. Although the sheriffs were amenable to this set-up, they informed us that we needed permission from the county administrator to set up the monitors on the public property. At first, the county administrator’s office was open to our plan but wanted to ensure we had properly considered public safety. We proposed a number of public safety strategies including protecting all the equipment, encasing it, erecting warning signs as well as signs explaining community air monitoring. Subsequently, the county administrator wanted to ensure that the project was self-insured so that the county would not be liable for any damages or loss of equipment. In addition to providing a letter from the university stating that the project was self-insured, the PI worked with the University’s Office of Supply Chain Management to develop a land use agreement with the county for the two weeks the monitor would be in place. After these assurances, the county administrator’s office wanted the university to waive subrogation for worker’s compensation, which would mean that if one of the researchers was hurt setting up the monitor on public property and the county was found liable for an on-site injury that the university would waive its right to recuperate worker’s compensation from the county. Such a waiver is against university policy.

After six weeks of negotiating, we were unable to secure permission to set up the monitor on public property between the library and the sheriff substation. With the county negotiation to set up the air monitors stalled; the research team set out to problem solve where to set up the air monitors. Working with team scientists, we were able to borrow a trailer to house the monitors. Borrowing a trailer meant that we could place the monitors in more areas since we did not need permission to set up monitors on a roof or on public property. Working with community partners who were also residents, we found a central location and a willing resident who let us draw power from their house for which we were able to provide renumeration. The monitors collected air samples to measure the particulate matter and VOCs. Without community partners who were also residents, we would most likely have had to further delay sampling and may not have been able to collect samples at all due to the impasse with the county.

Another major feature of doing transdisciplinary research is its ability to account for complexity. In addition to the importance of community organizations as research partners, it was necessary for academic researchers to transcend disciplinary boundaries to achieve study aims. For instance, through working across disciplines, we sought in this study to understand the problem of environmental stressors in its complexity—the cumulative impacts of environmental stressors and intersections with social disadvantaged on a small, rural community [17]. Not only did we investigate the problem in its complexity through collecting multiple kinds of data (i.e., water, air, and biological samples), we also sought to understand the residents’ environmental health concerns within the social context of ethnic, racial, and economic marginalization. Doing so was one example of what Higginbotham and colleagues (2001) consider the paradigmatics of transdisciplinary research—thinking and mapping possible interconnections of disciplinary knowledge about a problem [41]. Moreover, as Black and Black (2009) argue, transdisciplinary research is an integrative process such that researchers working together develop a shared framework to understand and extend discipline-specific theories, concepts and methods [10]. Following recent research into health disparities (e.g., [9,20,39]), this case study offers another example of utilizing both a transdisciplinary research and a CBPAR approach to understanding environmental health disparities to extend discipline-specific approaches to research.

As previously discussed, such community-engaged, transdisciplinary research as this is time-intensive in cultivating both relationships with community organizations and scientists from across the university (e.g., [7,29,36]). However, given the pressing social, environmental, and health challenges we face, such research is paramount to developing relevant and rigorous solutions to these kinds of pressing social and environmental problems [1,8,9]. One important feature of the current study was the openness of team members to different disciplinary perspectives and the willingness to work together to solve project-related problems and achieve the study’s aims. In addition to the challenges in doing this research, several important lessons were garnered from this study, with implications for doing transdisciplinary research.

### 4.3. Lessons Learned: Doing Transdisciplinary Research

Several important lessons were learned from the current case study on *how* to do community-driven, transdisciplinary, environmental health disparities research. First, social scientists have a key role to play as the “boundary spanner”, facilitating functional relationships across disciplines and communities. Second, consistent contact (e.g., monthly emails, conference calls, and in-person meetings) with both community partners and research scientists is necessary to build trust, increase buy-in, and facilitate progress on project aims throughout the lifespan of the project. Third, and similar to other research [27,30,37], an on-campus center, such as the EHSC, can be vital to establishing networks and connections, particularly in connecting junior faculty to scientists and researchers across multiple disciplines. The affiliated faculty of the on-campus center facilitated many collaborations across the transdisciplinary team. Moreover, center affiliates had an added incentive to work with pilot project PIs, since the center is rated by the federal funding agency (i.e., the NIEHS) on the funding pipeline it creates (i.e., the number of grants submitted to that agency). Thus, one of the goals of the center is to serve as an academic incubator. Scientists affiliated with the center tend to already have a deep investment in multi-disciplinary, community-engaged collaborations that encourages and supports such a study as this.

Additionally, successful transdisciplinary projects inform the organization and structure of the center itself, creating a positive feedback loop between transdisciplinary research and research centers. For instance, as Juarez and colleagues (2014) argue, a public health exposome model can bring together scientists who then, in turn, inform the center [20]. Public health exposome research combines insights from exposure science and social–ecological models to investigate multiple, underlying mechanisms of environmental exposure that may impact personal health, resulting in population level health disparities. Utilizing such an approach may inform the structure of an academic center through the need to bring together exposure scientists, policy analysts, and public health advocates. It is through the operationalization of this research paradigm that centers may structurally change in order to bring together transdisciplinary scholars and break down barriers generated by traditional disciplines [20].

The key challenges elaborated here, problem-solving, time-intensiveness and building trust, are also hallmark characteristics of transdisciplinary research (e.g., [8]). It is precisely through collective problem-solving and putting in the time necessary to solve emergent problems and build trust, that make transdisciplinary research possible. In addition to these challenges, there are also barriers to doing transdisciplinary scholarship. Many of these barriers coalesce around university structures and norms.

### 4.4. Barriers to Transdisciplinary Research

There are significant barriers to conducting transdisciplinary research. These barriers must be addressed in order to make this kind of research more accessible to researchers. One significant underlying structural tension lies in the disconnect between federal research funding priorities and how universities evaluate tenure-track professors for advancement. Although there is federal support for transdisciplinary research in the form of grant funding for PIs (e.g., multi-million dollar research driven grants, such as R-01 funding mechanism) and centers (e.g., funding for on-campus research centers that address certain problems such as those related to environmental health, i.e., P30 grant mechanism, or health disparities, i.e., P50 grant mechanism) there are significant university-level barriers, particularly for junior faculty. For example, the current academic reward structure for tenure-track positions, while resting on three primary areas, research, teaching and service, still primarily values research and, within research, peer-reviewed publications most of all. This means the rational utilitarian choice for junior faculty and doctoral candidates in today’s hyper-competitive academic job market (i.e., the steady decline in full-time faculty positions in the last four decades, [65]) is to conduct secondary data analysis on already existing datasets, in order to publish as quickly as possible, with rejection rates of around 75% and higher for many journals [66]. Although understood to be only a part of the time to publication (see [67]), the median review time, or time between the submission and acceptance of a paper at a single journal, is 100 days, according to a 2015 study of indexed papers in the PubMed database with listed submission and acceptance dates [68]. However, some scholars are finding that it may take about 9 months, after a series of rejections, for papers to find a journal home and come out to the public and academic audiences. [67]. Subsequently, junior faculty who engage in community-driven, transdisciplinary research are working against their own self-interest. Importantly, academic senates and academic affairs offices at universities have not developed or implemented an effective evaluation of the different and multiple contributions (e.g., openness to diverse and multiple perspectives; collaboration), in addition to publications, of transdisciplinary research. As such, universities are acting as institutional barriers to researchers doing transdisciplinary research.

There are several recommendations to reduce universities’ role as institutional barriers. The challenges of, and recommendations to overcome, these barriers are summarized in Table 3. First, the value, significance, rigor, and difficulty of transdisciplinary research must be better communicated to relevant, advising and decision-making bodies (departments, committees on academic personnel, external reviewers, deans, provosts, etc.) in order to deepen the understanding of the time-intensive nature of transdisciplinary research. Information on the rigors and contributions of transdisciplinary research can also be communicated to diverse fields through the continued development of practice-oriented journals, transdisciplinary journals, and special issues, which will help promote transdisciplinary research [32].

Another important platform for communicating the rigor, significance, and impact of transdisciplinary research is a department-prepared document of guidelines on evaluating dossiers for advancement. These documents could provide an executive summary, information on the program or department, as well as criteria for evaluation and exemplars of what a successful faculty member may do in the realms of research, teaching, and service. Such informative guidelines could then be submitted with dossiers to external letter writers and in-department reviewers, as well as college- and university-level review committees. For example, developing this understanding could follow from on-going work on advancing community-engaged and public scholarship [69]. In his seminal book, Boyer (1997) argues that there are four major foci of scholarship: discovery, integration, application and engagement, and teaching [70]. Scholarship of discovery is what is most commonly understood as products related to research and includes action research, experimental research, and ethnographical research, among other kinds of inquiry. Scholarship of integration includes types of research that aim to integrate knowledge across disciplinary perspectives, whereas, scholarships of application and engagement are models in which research attempts to address the outcomes of its application. Lastly, scholarship of teaching focuses on how research improves pedagogical knowledge and practices. In sum, there are examples, particularly in evaluating public scholarship, that could be adapted to inform colleagues, reviewers, and university decision-makers on the significance, impact, and contributions of transdisciplinary research.

In addition to informing faculty and other relevant decisionmakers of the rigor and importance of transdisciplinary research, another solution to university-level barriers is to change academic senate manuals, often referred to as the Academic Personnel Manual (APM) bylaws. Since the APM is the set of policies and procedures that govern and inform academic personnel, these rules must evolve to catch up to the practice of transdisciplinary approaches, including team science, convergence research, and collaborative scholarship that are advanced by federal funding priorities. This disconnect between federal funders and institutional evaluations creates a barrier for junior faculty and burgeoning graduate students, especially those who have been trained in these approaches. Updating performance evaluation also aligns universities with the cutting-edge research practices and funding priorities of federal funding agencies, such as the NIH and NSF in the U.S. As Team Science and Convergence Research are increasingly taught across graduate programs, the next generation of scholars will be better suited to federal funders’ priorities [71]. For instance, from 2006 to 2013, the number of multiple PI grants grew 15–20% of all major grants funded [72]. Changes to the APM to better account for and evaluate transdisciplinary research would go a long way in supporting and encouraging junior, midcareer, and senior faculty to participate in and do transdisciplinary work.

Finally, a more adequate approach to evaluating faculty records would reduce barriers to doing transdisciplinary research. As Stokols (2006) argues the necessary criteria for evaluating success of transdisciplinary research collaborations is the extent to which they encourage development of new conceptual models through empirical inquiries that integrate the theories and methods of particular fields (e.g., [29,72,73]). In order to meaningfully evaluate contributions, there are several emerging frameworks. One such framework is the CRediT taxonomy. This taxonomy, increasingly used in the biophysical natural sciences, includes 14 distinct roles that specify each contributor’s specific contribution to scientific output. This taxonomy can help what Allen and colleagues call a shift from authorship to contributorship [33,74]. Doing so can aid candidates in communicating their roles and decision-makers in understanding their contributions to transdisciplinary approaches to research (convergence research, team science, collaborative research, etc.). Such a taxonomy enables and empowers reviewers of professional advancement to understand the independence, significance, impact and rigor of one’s research, and their contributions to it.

Moreover, such taxonomies are probably a better measure of most academic research performance. Importantly, independence is necessary to the significance and impact of the candidate’s research in the field, and yet part of the significance and impact of one’s work relies on others—one’s students for collecting and analyzing data, advisors for training them, the communities they are working in, the departments that support them, and the universities that fund them. Many graduate students are trained to be collaborative through lab-models and working on PI projects but then they are evaluated as individuals, starting with the dissertation and moving throughout their academic careers. Independence of thought can be assessed in multiple ways, while acknowledging dependency on others to do meaningful science.

To advance transdisciplinary research, university-level changes are necessary. As others have argued, transdisciplinary research has the power to solve some of our most pressing problems (e.g., [2,4,20,27,36]). Junior faculty are particularly well situated to do this work given the changes in graduate training, particularly towards multi-disciplinary and interdisciplinary approaches, in the last thirty years [65]. Training the next generation of scholars also necessitates changes at the university-level to more adequately support these scholars as they continue their progress as independent researchers [7]. To train undergraduate and graduate students and not change the university structure would be a disservice to this future generation, and may hinder or delay solving our most pressing social problems.

## 5. Conclusions

Transdisciplinary research provides opportunities to advance knowledge to help solve our most pressing social, environmental, and health problems and to reduce disparities. Given this potential, major U.S. funders, such as the NIH and NSF, are encouraging and supporting of transdisciplinary research through multiple mechanisms. Transdisciplinary research can occur through many kinds of approaches including team science, convergence research, and collaborative scholarship. However, across these approaches, there are many challenges to doing this research, including managing different disciplinary perspectives, worldviews, and the time-intensity of working together to create novel knowledge. Adding the work and effort involved in the time-intensive process of community engaged research to this, and the complexity of the research process itself, can create multiple challenges on many fronts.

Changes to university policies and norms can reduce barriers and incentivize transdisciplinary research, particularly since such approaches are widely held to be necessary to solve our most pressing problems. Moreover, to better support the next generation of scholars being trained in these approaches, university-level changes, specifically in cultivating a better understanding and evaluation of transdisciplinary scholarship for its advancement, are necessary to reduce barriers and promote transdisciplinary scholarship. Currently, there is a university-shaped gap between researchers and federal funding agencies regarding the importance and necessity of transdisciplinary research. This gap will only become starker, as collaboration in peer-reviewed journals is increasing while solo authorship is decreasing [65].

Among the other approaches to transdisciplinary health sciences research described here, including translational, public health exposome, Team Science, One Health, and adaptive and participatory approaches, community-engaged transdisciplinary action research is an important line of inquiry for addressing environmental health disparities. This case study has provided some concrete ways on *how* to do this kind of research. The insights derived from this study, including a monthly research update, regularly scheduled conference calls and in-person meetings, and leveraging federally funded on-campus research centers to network and build trust across a large, multi-disciplinary research team are important strategies to *doing* transdisciplinary research. Having a boundary-spanner in the social sciences helped to bridge the divide between the community partners and multiple scientific disciplines necessary to achieve study aims. Working together within and across community partners, social sciences, biophysical sciences, and engineering, the research team was able to traverse these disciplines to create transdisciplinary research to answer community residents’ concerns regarding their environmental health. A convergence of research approaches is needed to provide the science necessary to better inform policymakers, empower community advocacy organizations, and add to new forms of knowledge.

## Figures and Tables

**Figure 1 ijerph-17-02303-f001:**
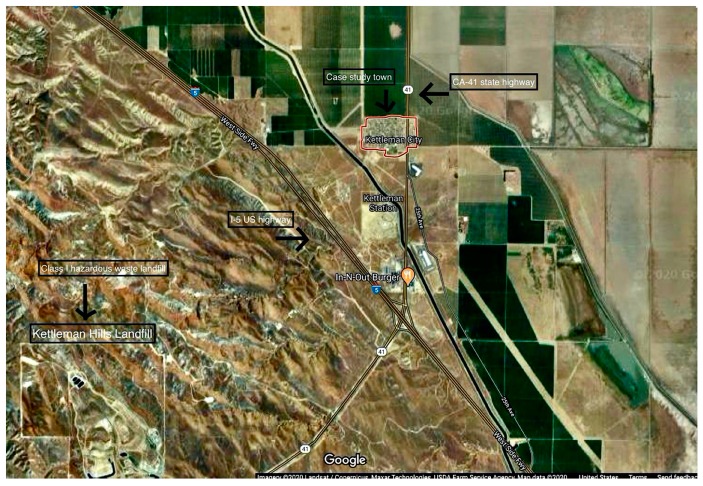
Satellite imagery of Kettleman City showing the community, two intersecting highways surrounded by agricultural land and hazardous waste landfill, Kettleman Hills (imagery from Landsat/Copernicus, Mazar Technologies, U.S. Department of Agriculture Farm Service Agency (2020) accessed via Google Maps at a scale of 2000 feet).

**Table 2 ijerph-17-02303-t002:** Potential environmental stressors, sample type, and analytes of the case study.

Potential environmental stressors	Sample type	Analytes
Diesel trucks (traffic)	Air quality monitoring	Particulate Matter (PM) and volatile organic compounds (VOCs)
Chlorine by-products	Household water for cohort	Trihalomethanes (THMs), haloacetic acids (HAAs)
Landfill runoff, construction, agriculture	Serum/plasma for cohort	Polychlorinated biphenyls (PCBs)
Trucks, landfill, pipes, agriculture, highway	Environmental health survey	Sociodemographics, perceptions of environmental health risks

**Table 3 ijerph-17-02303-t003:** Summary of institutional barriers to doing transdisciplinary research with recommendations to reduce these barriers.

Level of Institutional Barrier	Institutional Barrier	Recommendation to Reduce Barrier
Departments, colleges, university, discipline	Lack of understanding of the value, significance, rigor, and difficulty of transdisciplinary research across a range of evaluators	Increased communication of value, significance, rigor, and difficulty of transdisciplinary research to relevant decisionmakers (i.e., departments, committees on academic personnel, external reviewers, deans, provosts)
Discipline	Lack of specific outlets to promoting, sharing, and describing transdisciplinary research processes and findings	Development of practice-oriented journals, transdisciplinary journals, and special issues of journals
Departments, colleges	Lack of communication of evaluation criteria for transdisciplinary research	Department prepared guidelines with evaluation criteria and examples that could follow examples of community-engaged and public scholarship contributions to knowledge
University	Lack of policies and procedures that adequately takes into account recent changes in research activities	Update academic senate manuals (i.e., Academic Personnel Manuals) personnel manuals to provide guidance to more meaningfully evaluate transdisciplinary scholarship
Department, college, university, discipline	Lack of evaluation criteria for team science, convergence research, and collaborative scholarship	Use of new evaluation tools such as CRediT taxonomy to account for work contributed to collaborative research

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
