# Peer review of "Towards Convergence: How to Do Transdisciplinary Environmental Health Disparities Research"

_ijerph, 2020, doi:10.3390/ijerph17072303_

Round 1

Reviewer 1 Report

This work presents a review of the latest transdisciplinary methodologies related to environmental research and public health; also, based on the literature reviewed, a study case is carried out and the obtained results are discussed. In general, the work is interesting and well-written; yet, some issues have to be addressed for clarity purposes.

In line 31, ten works are referenced in group. Please discuss them in a more detailed way, mainly works presented in [6,7,9].

Statement in line 106 “there are seven key elements...” is yours? How do you determine these seven key elements? Are there others? Table 1 is mentioned in page 10 but it appears in page 3, please check, please add a sentence to indicate what Table 1 summarizes in page 3. The works reviewed to detect the key elements range from 2005 to 2015, can this section be updated (adding a new key element)?

For readers, please add labels (and arrows) to indicate relevant information in Figure 1.

More information about the 86-item survey (responses and questions) should be included as tables or graphs.

In section 4, it is not clear how do you use/implement the seven key elements presented in Table 1? Please discuss and highlight them.

Please include a Table that presents in a summarized way the institutional barriers to doing transdisciplinary research with the specific recommendations to reduce these barriers.

Could you include a flowchart about the steps of how to do transdisciplinary research?

Please update the reference list, if possible.

Please describe all the acronyms, e.g., NIH and NSF (abstract section), PI, etc.

Reviewer 2 Report

The Author presents a very original and scientifically interesting paper. I don't have any concern about the content of the manuscript and the very well documented issue.

I  wonder if the Author could shorten both the Introduction and the Discussion that are too long and detailed making the manuscript hardly legible and hardly understandable in some parts.

Reviewer 3 Report

Review for paper no.759139

Towards convergence: An ontology of transdisciplinary methodologies in health disparities research

submitted for publication in the International Journal of Environmental Research and Public Health

The paper provides a literature review on challenges of transdisciplinary research in broadly understood public health, then illustrates some of them on the example of a pilot research project investigating into the impact on environment pollution on human health in a small rural community, to draw conclusions on necessity of transdisciplinary research and obstacles it faces in practice.

I enjoyed reading the paper, though due to numerous intricate sentences the task was not easy. With some surprise I learned that the US research (conducted by universities) faces problems similar to those observed in my environment. I think the paper is going to be of interest to the audience of the Journal, especially for junior researchers planning their projects and seeking both methodological and practical advice.

I would like ask the author to consider the following comments:

The title

…feels too gassy: I have not seen too much ontology nor methodology in the content. The strength of the paper lies in naming practical, down-to-earth problems faced while searching answers for complex questions by project teams that involve people of different backgrounds, this naming done on the basis of extensive rearing on the subject and own experience.

3.2. Assembling a transdisciplinary research team

This chapter continues to provide background information on the research project serving as a case. The description is chaotic and not focused on the team (as suggested by the title).

Only the lines 333-342 (the very beginning of the section) directly describe the team. Within the same paragraph, starting from line 343, the author presents the problems of air and water quality observed by the inhabitants that defined the scope of tests on chemical pollutants present in the town. Then comes a general remark on benefits of community engaged science (lines 353-356) and collaborative teams, followed, within the same paragraph, by a statement of using a sequential multi-method approach, mentioning its two steps: survey among the town inhabitants and then collecting samples for chemical tests. Therefore, what is the topic of the paragraph?

The next paragraph describes research methods: a survey and methods of collecting air and water samples. Just a remark: the survey response rate was surprisingly low, 10% of 300 households in the entire town approached one by one – could be better considering the participation of “organizations with extensive and long ties to the community”.

Lines 389-393 – awkward formulation, as if water samples were taken from people, not households (is a person’s gender and age relevant for THM content in tap water?). I GUESS these people agreed to give blood samples – which is described in the next paragraph.

In my opinion, Lines 399-405 should be removed (nothing to do with the research team – they do not belong to the subsection) or, as they introduce Section 4, placed there.

Section 4.1. Keys to success

I do not understand the first sentence (line 408-409). The abbreviation PI has not been explained.

I cannot agree that the fact that the project leader was a social scientist has anything to do with the project success in terms of making different people work efficiently together. The research discipline is not important – the sense of purpose, stamina, managerial talent, “people skills” and enthusiasm are.

4.2. Elements of TD Research…

A small comment on the possible intake of the paper by non-US researchers - and the comical effect of illustrating the collaborative problem solving by the example of selecting location for the air monitoring system. Many would actually START from asking someone from the “organizations with extensive and long ties to the community” to put the thing in their aunt’s back yard and solve the problem in half an hour. However, maybe such a direct approach was out of question for cultural reasons (such as mistrust and hostility of the locals towards the research team) – if the example is to serve its purpose, some explanation is worth adding.

4.3. Lessons Learned… and 4.4. Barriers…

The paper is intended for an international journal. Please consider that reference to US research programs, grants, procedures, and organizations by acronyms is not very informative (even if the acronyms are explained). A more descriptive form would be welcome. For instance, what is the EHSC research center: a place in a particular university, or IT infrastructure to support research work? P30 grants, R-01 large scale grants say nothing to me. Nevertheless, the organizational barriers (e.g. rigid division into disciplines vs. project funding, university publishing policies that not support multiple-authored papers due to construction of researcher performance measurement systems, journals focusing on narrow disciplines) are commonly observed also outside US. Thus, the observations would be of interest to wide international audience.

Round 2

Reviewer 1 Report

All my comments and concerns have been properly addressed. I recommend the manuscript acceptance in its present form.